# From Broad Recall to Exact Distinction: Adversarial Curriculum Learning for Knowledge-Based VQA

## Abstract

Knowledge-based Visual Question Answering (KBVQA) aims to answer image-related questions by retrieving relevant facts from an external knowledge base, making the accuracy of knowledge retrieval crucial. However, a dominant bottleneck in existing systems is that inaccurate facts are fed to the answer generator. This issue stems from two key deficiencies: (i) an initial retrieval stage that relies on global visual features, often overlooking fine-grained evidence, and (ii) a reranking stage that lacks the ability to differentiate between confusing candidates, making the correct answer a lower priority. To address this, we propose the **Adv**ersarial **C**urriculum **L**earning (**Adv-CL**) framework, which tackles these two challenges sequentially. First, we design a Query-guided Multi-grained Recalling (QMR) strategy that leverages both global and query-guided local features to improve the recall quality and provide a diverse set of challenging negatives for reranker training. Subsequently, to enable exact distinction, we introduce an Adversarial Reranker Training (ART) paradigm, which compels the reranker to discern fine-grained distinctions among highly similar candidates. It employs a minimax game where a modulator network acts as an adversary against the reranker, dynamically creating a curriculum of hard negatives by up-weighting candidates that most confuse the reranker. This forces the model to develop its discriminative capability. In addition, we further introduce a Guarded Answer Generation (GAG) mechanism to mitigate the risk of retrieval failure exacerbating the system hallucination. Extensive experiments on public knowledge-based VQA benchmarks show that our method achieves state-of-the-art performance, validating the effectiveness and synergistic effect of broad recall and exact distinction.

## 1 Introduction

Visual Question Answering (VQA) aims to answer questions based on visual context. In recent years, multimodal large language models have made significant strides in this area Sun et al. (2024); Han et al.; Tschannen et al. (2025); Xiao et al. (2024). However, when confronted with knowledge-intensive queries involving domain-specific facts or rare entities, visual context alone is often insufficient. To address this, knowledge-based visual question answering (KBVQA) emerged Marino et al. (2021), which incorporates external knowledge bases to supplement visual information. This introduces a new, critical challenge: how to accurately retrieve relevant facts from a vast knowledge base to generate precise answers.

Although existing KBVQA systems, often built upon the Retrieval-Augmented Generation (RAG) framework, have achieved encouraging performance, their primary bottleneck is the provision of inaccurate facts to the answer generator. This issue stems from two key, sequential deficiencies in the retrieval pipeline: an initial recall stage that overlooks fine-grained evidence and a reranking stage that lacks discriminative power.

First, the initial retrieval quality is often suboptimal. Existing methods (Yan & Xie, 2024; Qi et al., 2024; Cocchi et al., 2025) typically encode images into a single global embedding. While capturing general context, this approach struggles to focus on the fine-grained local regions or objects essential for answering the query. For instance, determining a laptop's brand might depend on a minute logo,

a query-guided detail easily lost in a global representation. This reliance on coarse features results in a low recall ceiling, where crucial evidence is often omitted from the initial candidate pool, and also provides a noisy set of candidates for the subsequent stage.

Second, even when the correct fact is successfully recalled, the reranker often fails to distinguish it from a set of semantically similar but incorrect candidates. This is due to a lack of fine-grained discriminative power. Many approaches employ contrastive learning Khosla et al. (2020); Tian et al. (2020); Chuang et al. (2020) for reranker training. However, they often rely on static or non-adaptive negative sampling strategies. As the reranker's discriminative ability improves during training, it requires more challenging examples. The lack of dynamic, hard negatives leads to the learning signal being dominated by simple negatives, ultimately weakening the model's ability to make exact distinctions. Furthermore, a critical flaw in existing systems is the assumption that the generator must answer using the retrieved knowledge, which can lead to high-confidence hallucinations when the knowledge is erroneous.

Based on this analysis, we propose the Adversarial Curriculum Learning (**Adv-CL**) framework, a synergistic approach that tackles these two challenges sequentially. It comprises three core components: Query-guided Multi-grained Recalling (QMR), Adversarial Reranker Training (ART), and Guarded Answer Generation (GAG). Specifically, QMR leverages both global and query-guided local features to raise the recall ceiling. By identifying and emphasizing fine-grained, query-relevant regions, it provides a more comprehensive and diverse set of candidates for reranking. ART compels the reranker to discern fine-grained distinctions among highly similar candidates. It employs a minimax game where a modulator network acts as an adversary, dynamically creating a curriculum of hard negatives by up-weighting candidates that most confuse the reranker. This forces the model to develop precise discriminative abilities. GAG mitigates the risk of hallucination. It introduces two simple yet effective safeguards, *i.e.*, the prompt-based inspection and the retrieval discriminator, enabling the system to abstain from answering when retrieved knowledge is unreliable.

Extensive experiments on public KBVQA benchmarks demonstrate that our proposed Adv-CL framework achieves state-of-the-art performance, validating its effectiveness. In summary, our contributions are as follows: (i) We systematically identify and analyze a critical bottleneck in RAG-based KBVQA systems: inaccurate knowledge provision caused by suboptimal recall and an inability to distinguish between fine-grained candidates during reranking. (ii) We propose QMR to raise the recall ceiling and ART, which creates a dynamic curriculum of hard negatives to enhance the reranker's discriminative power. (iii) We introduce GAG, a mechanism that allows the model to refuse to answer when faced with unreliable retrieved knowledge, thus reducing hallucination.

## 2 RELATED WORK

### 2.1 KNOWLEDGE-BASED VQA

Unlike traditional VQA tasks, KBVQA requires the integration of external knowledge beyond the image content to answer questions. Recent datasets such as E-VQA Mensink et al. (2023b) and InfoSeek Chen et al. have further pushed the field by emphasizing fine-grained attribution of factual knowledge, introducing new challenges in multimodal reasoning.

Current approaches to KBVQA can be broadly categorized into three paradigms: (i) Jointly Optimized RAG Frameworks. This paradigm focuses on creating tight feedback loops between the retrieval and generation modules. For example, Hao et al. (2024) introduce a selector–answerer architecture where the generator provides pseudo-labels to iteratively refine knowledge selection. Similarly, Long et al. (2025) propose a reinforcement-based mechanism that uses feedback from the answer generator to directly optimize retrieval relevance. (ii) End-to-End Fine-tuned MLLMs. This approach integrates retrieval capabilities directly into the Multimodal Large Language Model (MLLM) via end-to-end training. Methods like Cocchi et al. (2025) and Zhang et al. (2024) incorporate self-reflective tokens, enabling models to autonomously assess the necessity of retrieval and the relevance of retrieved information. Others, such as Qi et al. (2024), focus on enhancing the MLLM's resilience to irrelevant information by introducing adversarial noise during training. (iii) Modular Training with Frozen LLMs. This paradigm keeps the pre-trained LLM frozen and concentrates on training lightweight modules for retrieval and reranking. For instance, Wang et al. (2024) and Weng et al. (2024) train a lightweight module to distill key information from knowledge

into soft prompts. Other works focus specifically on the retrieval and reranking components. Yan & Xie (2024) trains a Q-Former reranker on hard negatives to improve precision, Chen et al. (2025) develops a multi-modal encoder for initial retrieval followed by a reranking step, and Liu et al. (2025) fine-tunes an MLLM with LoRA to act as a powerful yet efficient retriever and reranker.

While the joint optimization and end-to-end MLLM paradigms facilitate rich information flow and strong instruction alignment, they often incur substantial computational costs and face challenges in scalability. In contrast, the modular paradigm offers greater efficiency and flexibility, allowing for easier integration of upgraded components Liu et al. (2025); Chen et al. (2025). Following this promising direction, our work focuses on designing a retrieval-reranking pipeline that can empower any off-the-shelf frozen LLM, addressing the core challenge of precise evidence identification.

### 2.2 HARD NEGATIVE MINING

The efficacy of contrastive learning is heavily dependent on the quality of negative samples. The strategies have evolved from simple in-batch negatives, which treat other positive samples within a mini-batch as negatives Yih et al. (2011); Henderson et al. (2017), to more sophisticated hard negative mining. This latter strategy involves deliberately selecting samples that are semantically close to the positive query but are incorrect, thereby compelling the model to learn fine-grained distinctions Robinson et al. (2020); Xia et al. (2021); Bucher et al. (2016). Recent efforts have sought to refine this process. For instance, Moreira et al. (2024) proposed a method to mitigate the risk of false negatives (incorrectly labeling a true positive as negative), while Zhang et al. (2025) adopted a two-stage strategy that bootstraps with random negatives before refining with hard ones. To manage computational overhead, methods like Yan & Xie (2024) resort to random sub-sampling from a larger pool of retrieved hard candidates. However, a common limitation across these approaches is their static nature. They typically rely on a pre-defined strategy or a fixed pool of candidates, failing to adapt as the model's discriminative power evolves during training. Our approach directly addresses this gap by dynamically adapting the difficulty of negatives to the model's current state, ensuring a persistent and effective learning signal throughout the training process.

### 2.3 CURRICULUM LEARNING

Inspired by human cognition, Curriculum Learning (CL) is a training strategy that improves model performance and convergence stability by presenting training examples in a meaningful order, typically from easy to difficult Bengio et al. (2009); Soviany et al. (2022); Wang et al. (2021). This paradigm has evolved significantly over time. Early approaches often relied on manually designed heuristics for difficulty, such as sentence length or concept frequency Platanios et al. (2019); Spitkovsky et al. (2010). More recent work has shifted towards automated methods for curriculum generation, using techniques like self-paced learning Kumar et al. (2010); Meng et al. (2017), transfer from teacher models Zhang et al. (2018); Zhou et al. (2020), and reinforcement learning Graves et al. (2017); Kumar et al. (2019). Our adversarial reranker training instantiates the curriculum learning paradigm, leveraging a min-max game mechanism to dynamically schedule training from easy to difficult cases for stable convergence and robust performance.

## 3 OBSERVATION

To elucidate the necessity of our proposed method, we conduct three diagnostic experiments to analyze the bottlenecks in typical KBVQA systems. The results reveal a coherent chain of deficiencies: a severe gap between retrieval quality and generation potential, the rapid decay of training signals in contrastive reranker training, and the vulnerability of static RAG systems to factual contamination from erroneous retrieval.

**O1. The Retrieval-Generation Capability Gap.** A key diagnostic for any RAG-based system is to isolate the primary performance bottleneck: the retriever or the generator. To quantify this, we evaluate several powerful generators (Mistral-7B, Llama3-8B, Qwen2.5-7B) on the E-VQA dataset under two conditions: (i) using perfect "oracle" ground-truth knowledge, and (ii) using knowledge retrieved by a powerful model. The results are stark. Under oracle conditions, the generators achieved near-perfect accuracies (91.2%, 90.4%, 89.4%), approaching the human consistency upper bound. However, when fed with retrieved knowledge, their accuracy plummeted to an average

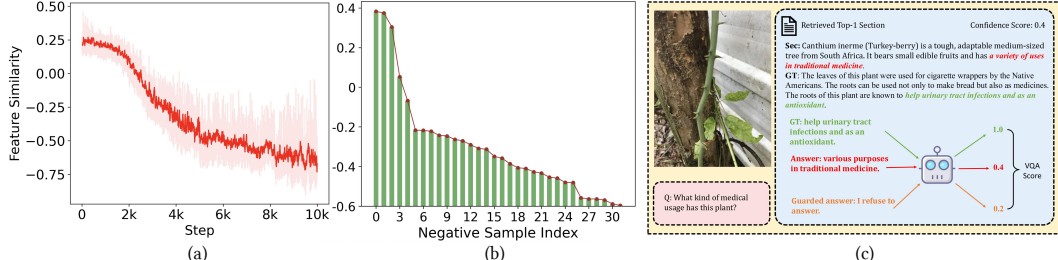

Figure 1: (a) The evolution of the similarity between negative samples and the query throughout training iterations. (b) Distribution of query-negative similarity scores. (c) The case of factual contamination in generation.

of 41.5%, a staggering drop of nearly 50 percentage points. This massive, consistent performance gap across all generators unequivocally identifies retrieval quality, not generation capacity, as the primary bottleneck in current KBVQA systems. This finding highlights the critical need to improve the precision of retrieved evidence.

**O2. Contrastive Learning Signal Decay.** Next, we investigate the training dynamics of the reranking stage by analyzing the InfoNCE Oord et al. (2018) loss contribution and query-negative similarity for 2,000 samples. We observe that conventional negative sampling strategies are suboptimal. As shown in Fig. 1(a), training a reranker with a static hard negative pool leads to a monotonic decline in loss, but this is a false indicator of progress. After approximately 2,500 steps, the average negative sample weights collapse, indicating the model has effectively identified these static negatives and its training has hit a premature plateau. Alternatively, relying on random top-k sampling is also inefficient. As seen in Fig. 1(b), roughly 40% of the top-ranked negatives in early training stages exhibit significant semantic deviation from the query, rendering them too easy to provide a useful learning signal. These findings confirm that a dynamic, adaptive strategy is required to continuously challenge the reranker.

**O3. Factual Contamination in Static RAG Systems.** Finally, we identify a critical vulnerability in static RAG systems, a phenomenon we term Factual Contamination. We compare a powerful MLLM's (Qwen2.5-VL) performance on E-VQA validation set under three settings: (i) with ground-truth knowledge, (ii) with incorrectly retrieved knowledge, and (iii) with no retrieval augmentation. The VQA scores were 84%, 18%, and 25%, respectively. The striking insight is that providing incorrect knowledge is significantly more detrimental than providing no knowledge at all. Plausible but irrelevant facts contaminate the model's reasoning process, inducing severe hallucinations where key attributes are swapped, producing answers that seem compelling but are factually wrong (see Fig. 1(c)). This underscores the need for a safeguard mechanism that empowers generator to recognize unreliable knowledge and refuse to answer, rather than propagating retrieval errors.

## 4 METHODOLOGY

The overall architecture of our proposed method, Adv-CL, is illustrated in Fig. 2. This section starts with the problem definition of KBVQA, followed by the details of the Query-guided Multi-grained Recalling (QMR) module in Sec. 4.1, the Adversarial Re-ranker Training (ART) strategy in Sec. 4.2, and finally the Guarded Answer Generation (GAG) in Sec. 4.3.

**Preliminaries.** For the KBVQA task, given an input image $I$ and a natural language question $Q$ about the image, the system is expected to generate a textual response $y$. A knowledge base $K$ is utilized during retrieval and generation, consisting of candidate multimodal KB entries.

### 4.1 QUERY-GUIDED MULTI-GRAINED RECALLING

**Query-guided Feature Aggregation.** To achieve comprehensive multi-grained retrieval, our approach extends beyond the global image features common in prior work by further incorporating pivotal local features. To this end, we design a query-guided feature aggregation module that selectively emphasizes and prioritizes relevant image patches. Specifically, we employ a pre-trained Vision-Language Model (VLM) with fine-grained image-text alignment capabilities Xiao et al. (2024),

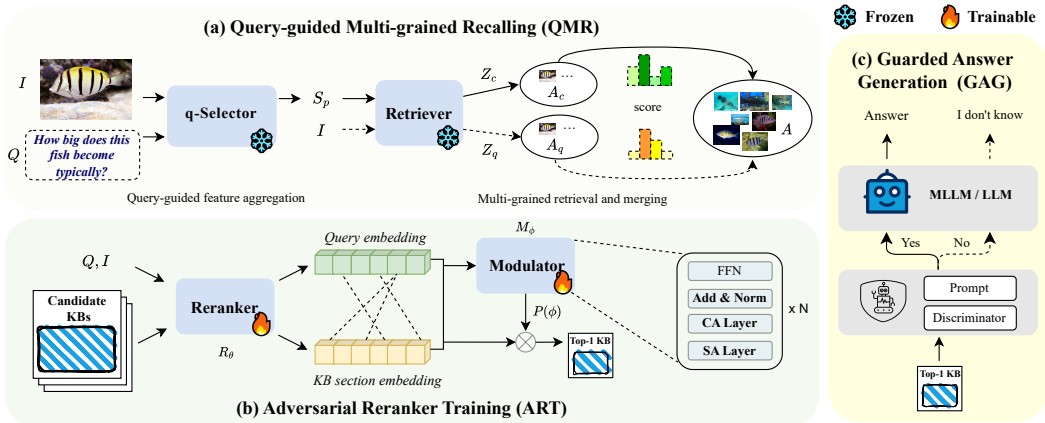

Figure 2: **Overview of the Adv-CL framework.** It involves three components: QMR retrieves candidate knowledge using multi-grained image features, ART conducts dynamic adversarial curriculum learning, filtering out the most conducive section for the response before feeding it into the proposed GAG to produce reliable answers.

referred to as *q-Selector*, to compute the relevance scores between image patches and the user question. Given a user query $q$ consisting of an input image $I$ and a question $Q$, we first encode the input image using the same image encoder as the knowledge base, typically EVA-CLIP Sun et al. (2024), to obtain the CLS token $\boldsymbol{Z}_c$ and image patches $\boldsymbol{Z}_p$ embeddings. For $\boldsymbol{Z}_p$, we use the *q-Selector* to calculate the normalized relevance score $\boldsymbol{S}_p$ between each patch and the question. The image patches are then weighted according to their relevance scores to derive the fine-grained feature $\boldsymbol{Z}_q$. The calculation process is shown in Eq. 1.

$$\boldsymbol{Z}_c, \boldsymbol{Z}_p = \text{EVA-CLIP}(I), \ \boldsymbol{Z}_c \in \mathbb{R}^{1 \times D}, \ \boldsymbol{Z}_p \in \mathbb{R}^{L \times D},$$
$$\boldsymbol{S}_p = q\text{-}Selector(I, Q), \ \boldsymbol{Z}_q = \boldsymbol{S}_p \odot \boldsymbol{Z}_p, \ \boldsymbol{S}_p \in \mathbb{R}^{L \times 1}, \tag{1}$$

where $L$ is the sequence length, $D$ is the feature dimension, and $\odot$ denotes the element-wise product.

**Multi-grained Retrieval and Merging.** Leveraging these global and salient local features, we perform multi-recall over the knowledge base $K$. Specifically, we establish two separate pathways for retrieval: one uses the global image feature (i.e., the CLS token $\boldsymbol{Z}_c$), and the other utilizes the query-guided fine-grained feature $\boldsymbol{Z}_q$. Both pathways leverage FAISS library Douze et al. (2024) for efficient similarity search, which employs a non-parametric function to compute the cosine similarity between the embedding of the image and all search indexes in the knowledge base $K$. Since the two retrieval processes are data-independent and share identical computational complexities, they can be fully parallelized. We denote the top-K KB entries corresponding to the images retrieved by the global features as $\mathbf{A}_c$, and those retrieved by the fine-grained features as $\mathbf{A}_q$. The final retrieved candidate pool is merged by $\mathbf{A} = \mathbf{A}_c \cup \mathbf{A}_q$.

In practice, we observe a high overlap rate between $\mathbf{A}_c$ and $\mathbf{A}_q$, which is expected because both the global and local features are derived from the same input and thus have similar feature distributions. However, the final recall results indicate that $\mathbf{A}_q$ still serves as an effective supplement to $\mathbf{A}_c$, as shown in Tab. D. The multi-grained retrieval and merging process facilitates the retrieval of samples at varying granularity and perspectives for the subsequent reranking stage. By retrieving and merging samples at multiple granularities and from diverse perspectives, our approach achieves better retrieval quality and ensures a more informative input for the subsequent reranking stage.

## 4.2 ADVERSARIAL RERANKER TRAINING

To better distinguish the candidate samples provided by QMR and mitigate the decaying gradient signals from negative samples during training, as discussed in Sec. 3, we propose an adversarial reranker training strategy based on the contrastive objective.

**Contrastive Reranking.** A standard reranker can be formulated as a model $\mathcal{R}_\theta$ that scores the relevance of the candidates given the user query $q$, generally a vision-language model. For a training

procedure consisting of the positive section $k_+$ and a set of negative sections $\mathbf{C}_{neg}$, the loss is calculated as:

$$\mathcal{L}_q = -log \frac{exp(s_+/\tau)}{exp(s_+/\tau) + \sum_{k_- \in \mathbf{C}_{neg}} exp(s_-/\tau)}, \quad (2)$$

where $s_+ = sim(\mathcal{R}_\theta(q), \mathcal{R}_\theta(k_+))$ and $s_- = sim(\mathcal{R}_\theta(q), \mathcal{R}_\theta(k_-))$ are the relevance scores, computed as the cosine similarity between their respective representations, and $\tau$ is a temperature hyperparameter. The negative samples are generally selected from in-batch samples within the same training batch Yih et al. (2011); Henderson et al. (2017), or offline-constructed, which are obtained based on the similarity between query anchors and candidate features Robinson et al. (2020); Xia et al. (2021); Bucher et al. (2016). Nevertheless, such negatives are sub-optimal for reranker training. As the reranker progresses, the model suffers from the rapid separation between negative sample representations and the positive ones, resulting in inefficient training and even stagnation.

**Adversarial Curriculum for Hard Negative Mining.** To overcome this limitation, we customize an adversarial training curriculum for the reranker by dynamically mining hard negative samples. During the adversarial training process of the reranker, the difficulty of negative samples is progressively increased, thereby ensuring a sustained and challenging learning signal. We frame the training process as a minimax game between two models: (i) the reranker ($\mathcal{R}_\theta$), which aims to learn discriminative representations by minimizing the contrastive loss. (ii) the modulator ($\mathcal{M}_\phi$), which aims to dynamically allocate "importance scores" to negative samples within a limited budget to maximize the contrastive loss. The overall objective is a minimax game formulated as:

$$\min_\theta \max_\phi \mathcal{L}_{adv}(\theta, \phi) = \mathbb{E}[-log \frac{exp(s_+^\theta/\tau)}{exp(s_+^\theta/\tau) + \sum_{k_- \in \mathbf{C}_{neg}} \mathbf{p}(\phi) \cdot exp(s_-^\theta/\tau)} + \lambda \cdot \mathcal{H}(\mathbf{p}(\phi))], \quad (3)$$

where $\mathbf{p}(\phi)$ represents the importance scores assigned to each negative sample according to the user query, $\mathcal{H}(\cdot)$ is an entropy regularization for the importance scores, and $\lambda$ is a balancing coefficient. We restrict the total budget of $\mathbf{p}(\phi)$ to the number of negative samples in each step. This means that the modulator must give greater importance to the more difficult negative samples and less importance to the easier ones. The entropy regularization term is aimed at promoting a diverse, non-degenerate weight distribution from the modulator, which prevents a collapse into the trivial strategy of exclusively selecting the hardest negative. The proposed minimax objective is solved by alternating updates to the reranker $\mathcal{R}_\theta$ and the modulator $\mathcal{M}_\phi$.

To facilitate understanding, we provide a detailed description of the implementation process. Firstly, we employ the reranker to extract the representation of the query and positive/negative samples. Next, we use the modulator to calculate an importance score for each negative sample: $\mathbf{p}(\phi) = \mathcal{M}_\phi(sg(\mathcal{R}_\theta(q)), sg(\mathcal{R}_\theta(k_-)))$, where $sg(\cdot)$ denotes the stop-gradient operator. The modulator $\mathcal{M}_\phi$ comprises a stack of standard transformer blocks, each containing a self-attention module, a cross-attention module, and a feed-forward network (FFN). To optimize the modulator and reranker alternately, we list their individual loss functions, as shown in Eq. 4 and Eq. 5.

$$\mathcal{L}_{\mathcal{M}_\phi} = log \frac{exp(sg(s_+^\theta)/\tau)}{exp(sg(s_+^\theta)/\tau) + \sum_{k_- \in \mathbf{C}_{neg}} \mathbf{p}(\phi) \cdot exp(sg(s_-^\theta)/\tau)} + \lambda \cdot \sum \mathbf{p}(\phi) \cdot log(\mathbf{p}(\phi)), \quad (4)$$

$$\mathcal{L}_{\mathcal{R}_\theta} = -log \frac{exp(s_+^\theta/\tau)}{exp(s_+^\theta/\tau) + \sum_{k_- \in \mathbf{C}_{neg}} sg(\mathbf{p}(\phi)) \cdot exp(s_-^\theta/\tau)}. \quad (5)$$

### 4.3 GUARDED ANSWER GENERATION

Although the proposed methodology improves retrieval and reranking, the inherent uncertainty in cross-modal retrieval can still lead to the selection of irrelevant or incorrect knowledge for the answer generator. To mitigate this risk, we propose two straightforward yet effective guarded mechanisms that allow the system to refrain from answering when retrieved knowledge is unreliable: (i) **Prompt-based inspection mechanism**: We instruct the MLLM to explicitly assess the reliability of the retrieved knowledge before answering. Specifically, we incorporate the designed prompt into the system prompt for robust generation, and the details can be found in Appendix A.2.3. While this zero-shot strategy yields consistent gains with no additional parameters, its effectiveness is bounded by the inherent capability of the generator. (ii) **Dedicated retrieval discriminator**: A small binary

Table 1: **Performance comparison on E-VQA and InfoSeek datasets.** Our Adv-CL framework outperforms all state-of-the-art KBVQA baselines across three different LLM generators, demonstrating its superior robustness and effectiveness.

| Method | GEN | RET | E-VQA | InfoSeek | | |
| --- | --- | --- | --- | --- | --- | --- |
| | | | Single-Hop | Unseen-Q | Unseen-E | ALL |
| BLIP-2 Li et al. (2023) | Flan-T5$_{XL}$ | N. | 12.6 | 12.7 | 12.3 | 12.5 |
| InstructBLIP Dai et al. (2023) | Flan-T5$_{XL}$ | N. | 11.9 | 8.9 | 7.4 | 8.1 |
| LLaVA-v1.5 Liu et al. (2024) | Vicuna-7B | N. | 16.3 | 9.6 | 9.4 | 9.5 |
| Qwen-2.5-VL Bai et al. (2025) | Qwen-2.5-7B | N. | 25.1 | – | – | 12.3 |
| DPR$_{V+T}$ Karpukhin et al. (2020) | BERT | Y. | 29.1 | – | – | 12.4 |
| RORA-VLM Qi et al. (2024) | Vicuna-7B | Y. | – | 27.3 | 25.1 | 26.2 |
| Wiki-LLaVA Caffagni et al. (2024) | Vicuna-7B | Y. | 21.8 | 27.8 | 28.9 | 28.4 |
| EchoSight Yan & Xie (2024) | Mistral-7B | Y. | 41.8 | – | – | 31.3 |
| ReflectiVA Cocchi et al. (2025) | LLaMA-3-8B | Y. | 35.5 | 28.6 | 28.1 | 28.3 |
| **Ours** | Mistral-7B | Y. | **46.0** | **33.9** | **34.2** | **34.0** |
| | LLaMa-3-8B | Y. | **46.5** | **34.1** | **33.8** | **33.9** |
| | Qwen-2.5-7B | Y. | **45.9** | **33.9** | **34.0** | **34.0** |

classifier, inserted before the decoding layer of the generator, predicts relevance from the last hidden states of the prefilling stage. Fully supervised training of this component enables superior performance, demonstrating particular effectiveness for large-scale models with limited parameters. More details can be found in Appendix A.2.3. By equipping the system with the ability to refuse rather than hallucinate, we take a critical step toward a trustworthy KBVQA system.

## 5 EXPERIMENTS

### 5.1 DATASETS AND METRICS

**Datasets.** Following recent methods Yan & Xie (2024); Cocchi et al. (2024); Caffagni et al. (2024); Qi et al. (2024), we evaluate our approach on two commonly used datasets, E-VQA Mensink et al. (2023a) and InfoSeek Chen et al. (2023), whose details are presented in Appendix A.1.1.

**Metrics.** We conduct a comprehensive evaluation of our proposed method from three critical perspectives: for **retrieval and reranking**, we utilize the standard Recall@K metric to assess whether the ground-truth knowledge is present among the top-K retrieved and reranked results and we evaluate our result on URL and section level, denoted as U and S, as detailed in Appendix A.1.2. For **visual question answering**, we report the VQA score, following conventional practice in the field. This score measures the holistic effectiveness of our system by calculating the accuracy of the generated answers against human-annotated ground-truth answers. For **answer reliability**, we define three metrics beyond standard accuracy (see Appendix A.1.3 for details): Abstention Precision (AP) measures the appropriateness of refusal when retrieval fails, Abstention Recall (AR) quantifies the detection rate of retrieval failures, and Valid Answer Rate (VAR) assesses answer accuracy conditioned on successful retrieval.

### 5.2 MAIN RESULTS

**Visual Question Answering.** The results of our method and other competitive baselines for the E-VQA and InfoSeek are shown in Tab. 1. And we have the following conclusions: (i) By adopting a retrieval-augmented generation framework and leveraging rich multimodal information from the knowledge base, the system achieves a substantial performance improvement. (ii) The retrieval-reranking architecture demonstrates superior effectiveness. Under the same retrieval encoder(EVA-CLIP-8B Sun et al. (2024)) and modality settings, the two-stage retrieval-reranking framework EchoSight Yan & Xie (2024) significantly outperforms other methods. This can be attributed to its decoupled objectives: the retrieval stage aims to obtain the candidate samples, while the reranking stage focuses on a precise selection. (iii) Our method achieves notable improvements in VQA score on both the E-VQA and InfoSeek datasets, surpassing the previous model by 3.2% and 3.0%,

respectively, without LLM fine-tuning. This performance gain stems from our proposed QMR and ART module, which effectively translates broad recall gains into enhanced final VQA accuracy.

**Knowledge Retrieval and Rerank.** We report the reranked recall of external knowledge in Tab. 2. Evaluated at the section level (see Appendix A.1.2 for rationale), Top-1 recall (R@1) on the E-VQA dataset increases from 29.0% to 32.3%, achieving an 11.4% gain. This improvement reflects more accurate top-rank positioning of relevant candidates, demonstrating the ability to filter superficially similar but semantically inconsistent samples through adversarial hard negative mining.

Table 2: **The result of the reranker recall against other methods.**

| Method | Metric | R@1 | R@5 |
|---|---|---|---|
| SeBe-VQA-Text | S | 20.0 | 37.5 |
| EchoSight | S | 29.0 | 44.0 |
| **Ours** | S | **32.3** | **44.6** |

## 5.3 ANALYSIS

### 5.3.1 ANALYSIS OF QMR

We analyze the effectiveness of query-guided multi-grained recalling by visualizing the relevance scores assigned by the *q-Selector* to image patches relative to the query question. As shown in Fig. 3, the original image is (a), and for the question "When was the building built in the given picture?", the *q-Selector* consistently attends to patches containing the building structure as (b), which is the most relevant region for answering the query. By emphasizing these corresponding patches and suppressing irrelevant areas, the method produces fine-grained representations that improve alignment between retrieved candidates and the query intent, thereby mitigating the impact of extraneous noise within the image on retrieval results.

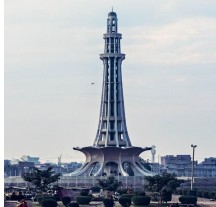 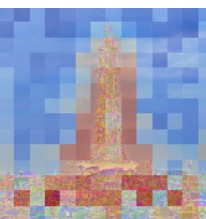

(a) Original $I$     (b) *q-Selector* scores

Figure 3: Visualization of the query-guided scores on the query image. Red color represents higher correlation.

### 5.3.2 ANALYSIS OF ART

In addition, we analyze the training process of the proposed adversarial reranker to understand its effectiveness. The loss curve (Fig. 4, left) shows an initial slight increase in contrastive loss, as the modulator upweights harder negatives and downweights easier ones. As training advances, the reranker gains dominance and the loss resumes a gradual decline. The interplay between the modulator and reranker is clearly illustrated in the right diagrams of Fig. 4. Following the application of ART, the scores of negative samples exhibit a more pronounced oscillatory decline, indicating improved discrimination of challenging samples. Furthermore, we visualize the heatmap of importance scores predicted by the modulator and the entropy of scores, as depicted in Fig. 5. Initially, the randomly initialized modulator yields a uniform score distribution with high entropy and no highlights. It then sharpens the distribution to raise the contrastive loss, resulting in visible highlights and a sharp entropy drop, reflecting active exploration. As the reranker strengthens, the modulator is increasingly challenged; guided by gradients, it elevates entropy again, leading to smoother scores and fewer highlights. The adversarial process converges when entropy is maximized—that is, scores follow a uniform distribution—marking the victory of the reranker.

## 5.4 ABLATION STUDY

### 5.4.1 ABLATION RESULTS ON QMR AND GAG.

We conducted ablation experiments on QMR and ART, with results and analysis presented in Appendix A.2.2 and A.2.3. Our findings confirm the importance of both components: QMR produces diverse, multi-perspective retrieval results, laying a solid foundation for reranking, while GAG improves the robustness of the KBVQA system through its retrieval discrimination mechanism.

### 5.4.2 ABLATION ON ART

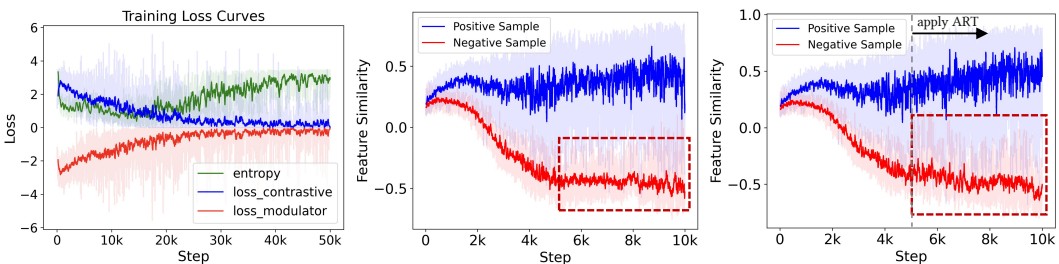

Figure 4: Training loss curves and the impact of ART on the sample similarity score.

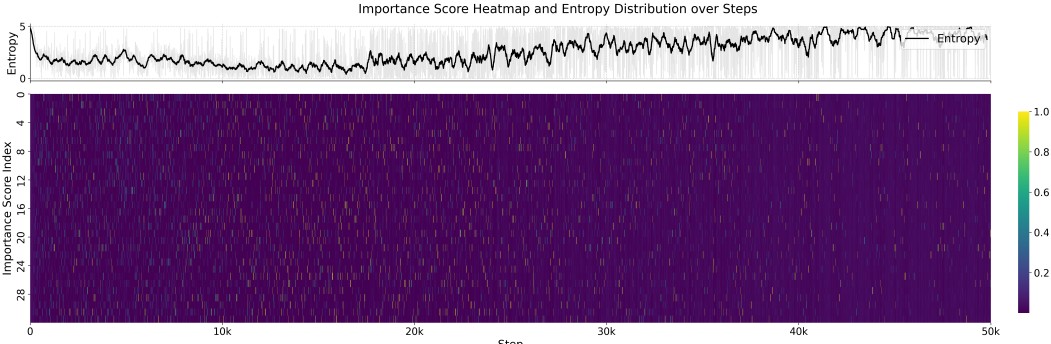

Figure 5: The heatmap of importance scores predicted by the modulator and the corresponding entropy curve. At the beginning of training, the modulator is randomly initialized with high entropy and a uniform heatmap. Then, entropy decreases, and the heatmap shows many highlights. With further training, entropy gradually increases, highlights decrease, and the system converges.

We also conduct an ablation study to validate the effectiveness of ART, as shown in Tab. 3. The reranker trained with ART achieves higher reranking recall within the same training recipe, which in turn enables the system to obtain higher accuracy. To further illustrate the effectiveness of ART, we plot the similarity scores between training samples and query features throughout the training process

Table 3: Ablation on ART.

| Method | R@1 | VQA Score |
|---|---|---|
| w/o ART | 29.3 | 41.8 |
| w/ ART | **32.2** | **42.8** |

under the settings with and without ART, as shown in Fig. 4. ART is activated at the position marked by the dotted line in the right figure. As the reranker trains, the similarity scores for negative samples gradually decrease, while the scores for positive samples slowly increase, which demonstrates the growing discriminative ability of the reranker. When ART is applied, the similarity scores of negative samples are weighted by the modulator. To ensure equivalent comparison, we calculate the log-average of the exponential similarity scores weighted by the modulator's predicted weights. We highlight the key differences between the two figures with dashed-line boxes. The application of ART results in higher average similarity scores for negative samples, indicating that the modulator assigns more weight to more challenging negative samples. Consequently, the reranker is trained to distinguish more difficult negative samples, thereby improving training efficiency.

## 6 CONCLUSION

In this paper, we address the dual bottlenecks within the retrieval–reranking module of the KBVQA system, which result in erroneous facts being passed to the generator. To tackle these challenges, we propose a novel framework, Adversarial Curriculum Learning (Adv-CL). Adv-CL first employs QMR to improve the quality of recalled candidates by integrating global features with query-guided local features. Then, ART employs a novel min-max game that creates a dynamic curriculum of hard negatives to hone the reranker's discriminative ability. A final GAG module provides a crucial safeguard against retrieval errors. Comprehensive experiments on multiple public KBVQA benchmarks demonstrate that our Adv-CL framework achieves state-of-the-art performance, validating its effectiveness and generalizability.

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

# A APPENDIX

This appendix contains the following parts:

- **Detailed Statistics of the Datasets** (Appendix A.1.1). We provide detailed statistics of the publicly available experimental datasets.

- **Details of Metrics for Retrieval and Reranking** (Appendix A.1.2). We detail the evaluation metrics (URL matching and section matching) and justify their respective use in assessing retrieval and reranking performance.

- **Details of Metrics for Generation** (Appendix A.1.3). We describe the three metrics used to evaluate the guarded generator's ability to correctly answer or decline based on retrieval success.

- **Implementation Details** (Appendix A.1.4). We specify the experimental setup, including the base models, optimizer, learning rate, batch size, and training steps.

- **Details of GAG** (Appendix A.1.5). We provide the detail of GAG, including the prompt.

- **Additional Ablation Studies** (Appendix A.2.1, A.2.2, A.2.3 and A.2.4). We provide the global ablation study and cost analysis as well as additional ablation studies on the QMR and GAG components. Moreover, we investigate the sensitivity to the balancing coefficient $\lambda$.

- **Supplement to related work.** We provide a systematic review of other promising advances in KBVQA, including the approaches about OK-VQA and A-OKVQA.

- **Case Study: Tracing One Question Through QMR → ART → GAG** To demonstrate how QMR, ART, and GAG operate in concert, we trace the full pipeline on a single sample.

- **The Use of LLMs**. (Appendix A.5). We discuss the utilization of Large Language Models in our work.

- **Ethics and Reproducibility Statement**.(Appendix A.6). This section contains our statements regarding research ethics and the statement of reproducibility.

## A.1 EXPERIMENT SETTINGS

### A.1.1 DATASETS

The detailed statistics of our used dataset are shown in Tab. A.1.1.

**Encyclopedic-VQA** Mensink et al. (2023a) is a large-scale VQA dataset featuring visual questions about the fine-grained categories from iNaturalist 2021 Van Horn et al. (2021) and instances from Google Landmarks Dataset v2 Weyand et al. (2020). It contains 221K unique question-answer pairs, each of which is matched with up to 5 images, resulting in a total of 1M VQA samples. Moreover, the dataset is accompanied by a knowledge base derived from Wikipedia, consisting of visual images and text documents from Wikipedia. The questions are of four types: templated, single-hop questions, automatically generated single-hop questions, multi-answer questions, and two-hop questions. Our experiments on E-VQA only take into account templated and automated single-hop questions to evaluate the effectiveness of the model.

Table A: The details of experimental datasets, which are composed of E-VQA and InfoSeek.

| Dataset | Question Type | Number of IQA pairs | | |
|---|---|---|---|---|
| | | Train | Val | Test |
| **E-VQA** | Templated | 66,535 | 1,827 | 1,000 |
| | Automatic | 737,114 | 8,025 | 2,750 |
| | Multi Answer | 112,736 | 1,844 | 1,000 |
| | Total | 916,385 | 11,696 | 4,750 |
| **InfoSeek** | Total | 902,509 | – | 71,335 |

Table B: **Results of URL recall.** Our method achieves performance comparable to Echosight, and both outperform the other competitors.

| Dataset | Method | Metric | R@K | | | |
|---|---|---|---|---|---|---|
| | | | K=1 | K=5 | K=20 | K=40 |
| E-VQA | Wiki-LLaVA | U | 3.3 | 9.9 | 17.5 | – |
| | ReflectiVA | U | 15.6 | 36.1 | 49.8 | – |
| | EchoSight | U | 36.5 | 47.9 | 50.2 | 56.1 |
| | **Ours** | U | **36.7** | **48.3** | **51.1** | **58.4** |
| InfoSeek | Wiki-LLaVA | U | 36.9 | 66.1 | 78.4 | – |
| | RoRA-VLM | U | 29.6 | 41.4 | 46.6 | – |
| | EchoSight | U | 53.2 | 74.0 | 77.9 | 81.9 |
| | **Ours** | U | **54.2** | **74.6** | **79.6** | **82.8** |

**InfoSeek** is a VQA dataset tailored for information-seeking questions that require knowledge beyond common sense, including 1.3M visual information-seeking questions, covering more than 11K visual entities from OVEN Chen et al.. The evaluation set is divided into two subsets: Unseen Entity and Unseen Question. The Infoseek dataset consists of a training set and three evaluation sets: InfoSeek$_{wikidata}$, InfoSeek$_{Validation}$, and InfoSeek$_{Human}$. The training set, together with the first two evaluation sets, transforms knowledge triples in Wikidata into natural language questions, resulting in 1.3M examples. InfoSeek$_{Human}$ contains 8.9K samples annotated by humans to simulate real information-seeking intentions. We use the filtered 100K knowledge base from E-VQA as in the previous work Yan & Xie (2024). Since the InfoSeek dataset does not have a golden evidence section label, we conducted zero-shot experiments on the InfoSeek dataset to evaluate the model's performance. Following prior work for a consistent comparison, we use the validation split of InfoSeek as our test set.

### A.1.2 Details of Metric for Retrieval and Reranking

For evaluating the retriever, we employ URL matching recall as the metric, reflecting the retriever's ability to retrieve true articles.

However, for reranking, we adopt a section-level matching metric to evaluate whether the ground-truth section appears among the top-K retrieved sections, as opposed to URL-level matching. Since a single Wikipedia URL may contain multiple sections, URL matching tends to overestimate reranking performance: any section from the correct URL is considered a hit, even if it is not the actual evidence section. Such coarse-grained evaluation does not align with the finer granularity required in the reranking stage.

Tab. B presents the URL matching results, showing that our method performs comparably to Yan & Xie (2024). However, as shown in Tab. 2, our method achieves significantly higher recall@1 under section matching. This suggests that high URL recall does not guarantee superior section-level retrieval accuracy.

Therefore, we argue that section matching more accurately reflects the reranking model's ability to prioritize the true evidence section, justifying its use as the primary evaluation metric in our reranking experiments.

### A.1.3 Details of Metrics for Generation

Beyond standard accuracy, we assess reliability by analyzing retrieval-generation interactions across four cases: True Positive (TP): Retrieval succeeds, answer is correct. True Negative (TN): Retrieval succeeds, answer is incorrect. False Positive (FP): Retrieval fails, system correctly abstains. False Negative (FN): Retrieval fails, system hallucinates an answer. Based on these, we define: Abstention Precision (AP): Proportion of correct abstentions among all abstentions, measuring appropriate refusal upon retrieval failure while answering correctly upon success. Abstention Recall (AR): Proportion of retrieval failures correctly detected, evaluating system sensitivity to retrieval breakdowns. Valid Answer Rate (VAR): Answer accuracy when retrieval is successful, reflecting performance

under reliable knowledge provision.

$$AP = \frac{FP}{FP + TN_{refuse}}, \quad AR = \frac{FP}{FP + FN}, \quad VAR = \frac{TP}{TP + TN}, \quad \text{(A)}$$

where $TN_{refuse}$ indicates that the retrieval is successful but the system refuses to answer, which is a subset of TN.

### A.1.4 IMPLEMENTATION DETAILS

**Retrieval and Reranking.** For retrieval, we use the visual encoder of EVA-CLIP-8B Sun et al. (2024) to extract image features and build a knowledge base with the FAISS Douze et al. (2024) library for efficient image retrieval. For query-guided multi-grained retrieval, we employ the well-trained FLAIR Xiao et al. (2024) as the *q-Selector*. For reranking, following Yan & Xie (2024), we use BLIP-2 Li et al. (2023) as the reranker to extract features of multimodal queries and knowledge sections, and initialize a lightweight modulator with two standard Transformer blocks, which includes multi-head self-attention and cross-attention modules and an FFN module. We offline extract initial hard negatives using the proposed QMR strategy and randomly select 32 negative samples in each training step for adversarial learning. For the reranker, we use the OneCycleLR learning rate scheduler and AdamW optimizer Loshchilov & Hutter (2017) with a learning rate of 1e-4 and a batch size of 32. For the modulator, we use the AdamW optimizer with a constant learning rate of 5e-5 to ensure stable training. We first train the reranker for 50K steps to build its discriminative ability. Then, we activate the modulator for adversarial training, with a total of 150K steps.

**Answer Generation.** We employ both large language models (Llama3-8B Liu et al. (2024) and Qwen2.5-7B Qwen et al. (2025)) and multimodal large models (Qwen2.5-VL-7B Bai et al. (2025)) as generators, repectively.

### A.1.5 DETAILS OF GAG

We provide GAG with two optional strategies: a prompt-based inspection mechanism and a dedicated retrieval discriminator. For the prompt-based inspection, the prompt is as follows:

> **Prompt for Inspection Mechanism**
>
> <Query>\n<Context>
> Please first determine whether the provided context can answer the question posed for the image, and output it in the format of <think>Yes</think> or <think>No</think>.
> If it is <think>Yes</think>, it means that the correct answer to the question can be obtained based on the provided context, and the answer to the question is output in the format of <answer></answer>.
> If it is <think>No</think>, it means that the correct answer to the question cannot be obtained based on the provided context, and no other output is required.

For the retrieval discriminator, we sampled 20K training instances from the E-VQA dataset. Each instance consists of a <Query><Context> pair and a binary label of Yes or No. If the <Query> and <Context> form a matched positive pair according to the dataset annotations, the corresponding label is Yes; otherwise, it is No. To improve training efficiency, we selected negative samples for each <Query> exclusively from the hard set filtered by the retrieval module. Additionally, we balanced the ratio of samples corresponding to the two labels in the training data to prevent model bias. The definitions and computational details of AP/AR/VAR are provided in Appendix A.1.3; these metrics evaluate the joint performance of the retrieval and answering components in a RAG system. The final VQA accuracy is shown in the Tab. A.2.3. Note that our reported accuracy includes cases where the model correctly identifies retrieval failures.

## A.2 ADDITION ABLATION STUDY

### A.2.1 GLOBAL ABLATION AND EFFICIENCY ANALYSIS

Table C presents a comprehensive analysis of the computational overhead introduced by QMR, ART, and GAG in terms of GFLOPs, Training Speed (TSpeed), and Inference Latency.

| Components | | | Efficiency & Performance Metrics | | | |
|:---:|:---:|:---:|:---:|:---:|:---:|:---:|
| QMR | ART | GAG | GFLOPs | Train Speed (it/s) | Latency (ms) | VQA Score |
| - | - | - | 14,334 | 2.12 | 2,793 | 41.8 |
| ✓ | - | - | 14,360 | 2.12 | 3,744 | 42.3 |
| ✓ | ✓ | - | 14,360 | 1.90 | 3,744 | 46.0 |
| ✓ | ✓ | ✓ | 15,070 | 1.90 | 3,744 | 45.9 |
| - | ✓ | ✓ | 15,044 | 1.90 | 2,793 | 42.5 |
| ✓ | - | ✓ | 15,070 | 2.12 | 3,744 | 42.1 |

Table C: **Cost Analysis and Global Ablation Study.** We report the computational complexity (GFLOPs), Training Speed (iterations per second), Inference Latency (milliseconds), and the final VQA performance. Note that ART only affects training speed, while QMR primarily impacts inference latency.

**QMR:** The primary latency overhead in QMR arises from the computation of query-weighted features. However, this is mitigated by our system's parallel execution design. Unlike mainstream cross-modal methods that require a computationally expensive second encoding step, QMR leverages efficient image-to-image retrieval (powered by Faiss). Consequently, while QMR incurs a slight inference cost compared to pure image retrieval, its latency remains comparable to standard Image+Text paradigms. Crucially, this enables multi-granularity retrieval, providing high-quality negatives that substantially improve ART's contrastive training.

**ART:** Although ART introduces a regulator module, the associated training overhead is marginal. As shown in Table C, TSpeed decreases only slightly (from 2.12 to 1.90 it/s). This efficiency stems from the regulator's lightweight architecture (only two Transformer blocks) and the short input sequence length (limited to the batch size). More importantly, ART is exclusively a training-time optimization; it imposes zero additional cost during inference, presenting a highly favorable trade-off for the permanent performance gains it secures.

**GAG:** This module is designed as an optional branch during the RAG inference phase. It provides a conditional mechanism for users to handle refusal responses. Therefore, GAG imposes no mandatory latency penalty on the standard inference process.

### A.2.2 ABLATION RESULTS ON QMR.

To validate the necessity of the query guidance in local patch selection, we replace the *q-Selector* with (i) random sampling and (ii) pure visual saliency detection without query conditions (PVS), evaluating performance on Recall@20 and Recall@40, with results shown in Tab. D. Results consistently show that both blind selection strategies degrade performance by introducing an influx of semantically irrelevant areas. Random sampling suffers severely from background noise, in contrast, saliency detection is misled by the poor alignment between visual saliency and query-specific rel-

Table D: Ablation of QMR.

| Method | R@20 | R@40 |
|:---|:---:|:---:|
| **Random** | 30.7 | 35.7 |
| **PVS** | 36.9 | 43.5 |
| **Ours** | **51.8** | **56.8** |
| w/o QMR | 50.1 | 56.2 |
| w/o Global | 48.0 | 52.1 |

evance. An ablation study validates the necessity of query guidance local features for multi-grained visual recalling: using only global features yields a Recall@40 of 56.2, while local features alone degrade performance to 52.1. The integration of both within QMR demonstrates a clear synergistic effect, achieving the highest recall and establishing a solid foundation for subsequent reranking.

### A.2.3 ABLATION ON GAG

Finally, we conduct an ablation study on the proposed GAG, as shown in Tab. E. We employ the 3B/7B models of Qwen2.5-VL as the generator. When no retrieval discrimination mechanism is applied, the AP and AR metrics are zero, indicating that the generator will not refuse to answer regardless of whether the retrieval is successful. When the prompt-based inspection is applied, the system's AP and AR are 0.87/0.96 and 0.69/0.74, respectively, indicating that the generator refuses to answer those queries with failed retrieval.

However, the VAR decreases slightly, suggesting that the generator misjudges some queries with successful retrieval and also refuses to answer them. When we apply the retrieval discriminator, we first utilize the reranker to filter 10K query-knowledge pairs from the E-VQA training set, assigning each data point a binary label based on whether the knowledge is paired with the query. For

Table E: Ablation results on GAG. PI. represents the prompt-based inspection and RD. represents the retrieval discriminator.

| Method | Qwen2.5-VL-3B | | | | Qwen2.5-VL-7B | | | |
|---|---|---|---|---|---|---|---|---|
| | AP | AR | VAR | Acc | AP | AR | VAR | Acc |
| w/o GAG | 0 | 0 | **0.80** | 0.43 | 0 | 0 | **0.85** | 0.46 |
| w/ PI. | 0.87 | 0.69 | 0.64 | 0.75 | **0.96** | **0.74** | **0.79** | **0.81** |
| w/ RD. | **0.94** | **0.72** | 0.76 | **0.79** | 0.95 | **0.74** | 0.78 | **0.81** |

Qwen2.5-VL-3B, the retrieval discriminator achieves better performance than prompt-based inspection, but there is no advantage for Qwen2.5-VL-7B. This validates our opinion in Sec. 4.3 that the prompt-based inspection depends on the generator's capability. However, the retrieval discriminator incurs additional training costs, and its result depends on the quality of training data. In practice, we suggest prioritizing the cheaper prompt-based inspection to implement GAG.

### A.2.4 Ablation on balancing coefficient $\lambda$

We also perform an ablation experiment on the balancing coefficient $\lambda$ in Eq. 3. Experimental results demonstrate that, within a reasonable range, the recall is insensitive to the choice of $\lambda$. This parameter weights the entropy-loss term: a larger $\lambda$ encourages the predicted importance scores to approach a uniform distribution, whereas a smaller $\lambda$ may drive the model to focus exclusively on the hardest instances, sacrificing sample diversity. We evaluated $\lambda \in \{0.005, 0.01, 0.1, 0.5\}$ and observed that the re-ranking recall varied by less than 2%. Across all settings, entropy loss exhibited a consistent trend.

### A.3 Supplement to related work

Due to space limit, we only discuss the related works of EVQA and InfoSeek datasets in Sec. 2.1, leaving limited room to discuss other promising advances in KBVQA. To fill this gap, we provide a systematic review of these emerging directions as follows.

Early OK-VQA Marino et al. (2019) and A-OKVQA Schwenk et al. (2022) highlighted the importance of knowledge in VQA, but focused almost exclusively on commonsense knowledge. For instance, KAT Gui et al. (2022) and REVIVE Lin et al. (2022) prompt an LLM to generate answer candidates, while RA-VQA Lin & Byrne (2022) and prior work Qu et al. (2021); Gao et al. (2022) condition generation on knowledge retrieved from an external KB to boost VQA performance. FLMR Lin et al. (2023) fuses token-level visual and textual features into multi-dimensional embeddings to capture finer query–document relevance, and its successor PreFLMR Lin et al. (2024) scales pre-training to over ten million image–text pairs, yielding a powerful multimodal retriever. Self-bootstrapped Hao et al. (2024) further proposes a co-training scheme in which a selector picks key knowledge documents for the answerer and the answerer returns pseudo-labels to refine the selector, bootstrapping both components without extra annotations.

### A.4 Case Study: Tracing One Question Through QMR → ART → GAG

To demonstrate how QMR, ART, and GAG operate in concert, we trace the full pipeline on the question "When was this building built?", as demonstrated in Fig. A. (1) QMR. The global pathway retrieves six Wikipedia entries whose images share the overall museum style, ranking the correct article "Grassi Muzeum" at position 4. Meanwhile, the query-guided local pathway attends to the marginal inscription "Grassi Muzeum" visible on the façade, producing an additional three candidates. The merged pool contains the ground-truth section at position 2. (2) ART. The modulator assigns different importance scores to distinct negative samples, with more challenging negative samples receiving higher weights. (3) GAG. We emulate retrieval failure by forcing the incorrect "1950s" section into Top-1. Through our designed prompt-based inspection mechanism or dedicated retrieval discriminator, the generator recognises the irrelevance of the provided section. Consequently, the generator abstains with "I refuse to answer" instead of hallucinating the wrong year.

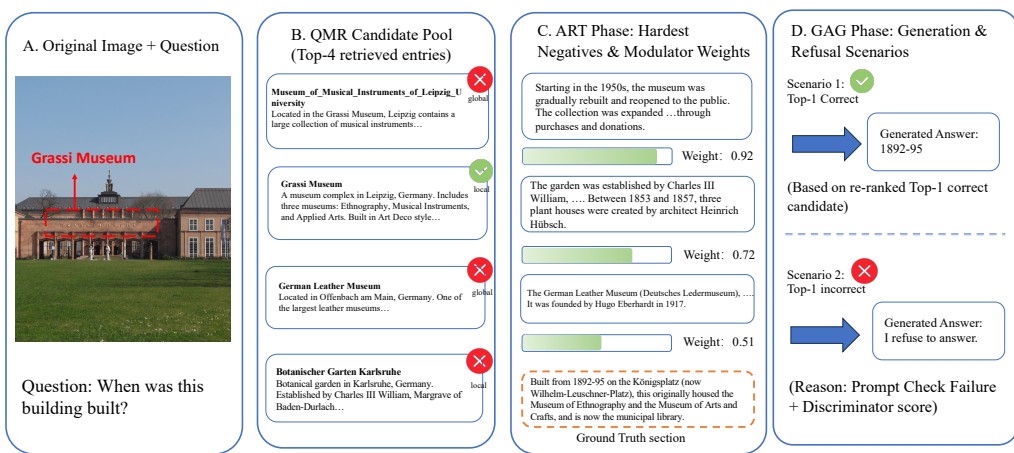

Figure A: Complete workflow of Adv-CL on a single example

## A.5 THE USE OF LLMs

In the preparation of this manuscript, we utilized a Large Language Model (LLM). The tool was employed solely for grammar checking and polishing the language expression. All scientific content, analysis, and conclusions remain entirely our own. The authors take full responsibility for the entire content of the paper.

## A.6 ETHICS AND REPRODUCIBILITY STATEMENT

This work complies with the ICLR Code of Ethics. We are not aware of significant ethical concerns arising from this research, which utilizes publicly available datasets and base models. Detailed experimental settings can be found in Appendix A.1.1 and A.1.4. Our experiments are conducted entirely on open-weight models. To ensure reproducibility, we will provide the full source code and model.

