# OpenReview forum: "From Broad Recall to Exact Distinction: Adversarial Curriculum Learning for Knowledge-Based VQA"
_ICLR.cc/2026/Conference — ICLR 2026 Conference Withdrawn Submission_

### Official Review · Reviewer_Nd2P · 2025-10-26

**Soundness:** 3
**Presentation:** 3
**Contribution:** 2
**Rating:** 4
**Confidence:** 4

**Summary:**

This paper proposes Adv-CL (Adversarial Curriculum Learning) for knowledge-based visual question answering (KBVQA).

Adv-CL includes three modules:
QMR (Query-guided Multi-grained Recalling): combines global and local query-guided features to improve recall.
ART (Adversarial Reranker Training): uses a modulator network in a minimax game to dynamically generate hard negatives.
GAG (Guarded Answer Generation): adds a check so the model can abstain when retrieved knowledge is unreliable.

On E-VQA and InfoSeek, Adv-CL achieves state-of-the-art results and improves both retrieval accuracy and answer reliability.

**Strengths:**

1. The adversarial curriculum dynamically adapts hard negatives and improves reranker learning.
2. Strong and consistent improvements on multiple benchmarks.
3. Clear visualizations of query-guided features and modulator behavior.

**Weaknesses:**

1. Each module builds on known techniques; the main innovation lies in the overall framework.
2. The joint contribution of QMR, ART, and GAG is not clearly separated. There should be a more clear table showing how each module contributes to the overall improvement
3. ART introduces an additional modulator transformer and adversarial optimization; the extra training cost and inference overhead (if any) are not clearly reported.
4. Only E-VQA and InfoSeek are tested. It would strengthen generalizability to include other KBVQA or open-domain RAG datasets (e.g., OK-VQA).
5. It seems that the literature review is not comprehensive. There are more works in OK-VQA (a popular KBVQA dataset as well) that should be mentioned and discussed.

**Questions:**

How sensitive is ART to the λ parameter in Eq. 3?

---

> ### Author Response · Authors · 2025-11-21
> **Response to Reviewer Nd2P**
>
> Thank you very much for the time you devoted and for your invaluable suggestions, from which we have benefited greatly. Below are our responses and supplementary information to your questions.
>
> >Q1:Innovation
>
> The core innovation lies in the adaptive curriculum of hard-negative generation within the ART mechanism. While the notion of hard negatives is not new, our experiments show that static sampling strategies fail to keep pace with training: the negatives do not automatically become harder, so the reranker’s supervisory signal dries up prematurely. We therefore introduce dynamic hard-negative mining that progressively raises the difficulty of negatives as training proceeds, continuously sharpening the reranker’s ability to discriminate true evidence.
> The use of adversarial learning for this mining process was derived from empirical observations, not a superficial stitching of ideas. Ablation studies on ART (Tab. 3 & Fig. 4) demonstrate that the regulator assigns larger weights to more challenging negatives, lifting the bar of “hardness” and enabling the reranker to master ever subtler distractors.
>
> >Q2 & Q3:Global ablation and cost analysis
>
> We measured the computation and latency introduced by QMR, ART, and GAG during training and inference and show the proposed module's effect on the final VQA score, as shown in Table 1. ART does raise training costs because of the extra regulator, but the overhead is tiny for two reasons. First, the regulator’s input sequence length equals the number of positives and negatives in one batch—typically only a few dozen—and the regulator itself is just two standard Transformer blocks, so the added reranker-training latency is minimal. Second, ART is used only at training time and adds zero inference cost; it is a one-time expense that permanently benefits the RAG system, which we regard as worthwhile.
>
> >Q4: Test on OK-VQA
>
> The early OK-VQA dataset introduced questions that demand **broad commonsense knowledge**. Such questions are increasingly covered by large-scale architectures (e.g., multilingual LLMs) during pre-training; their heavy reliance on parametric priors dilutes the measurable impact of an external knowledge base. Moreover, our retrieval pipeline performs image-to-image (I2I) search, which mandates a multimodal knowledge base, whereas OK-VQA provides only textual knowledge. The later introduction of E-VQA and InfoSeek brought truly multimodal knowledge repositories. Consequently, we adopt E-VQA and InfoSeek to evaluate our KB-VQA approach, aligning with the most recent literature.
>
> >Q5: Related work of OK-VQA and A-OKVQA:
>
> Thank you for pointing out the insufficient coverage of the literature review. The datasets used in our work mainly target recently proposed knowledge-intensive VQA tasks that require recognising fine-grained categories and detailed instance-level attributes. Nevertheless, to make the KBVQA section of the survey more complete, we have now summarised the relevant studies on OK-VQA and A-OKVQA as follows:
> Early OK-VQA and A-OKVQA highlighted the importance of knowledge in VQA, but focused almost exclusively on commonsense knowledge. For instance, KAT [1] and REVIVE [2] prompt an LLM to generate answer candidates, while RA-VQA [3] and prior work [4,5] condition generation on knowledge retrieved from an external KB to boost VQA performance. FLMR [6] fuses token-level visual and textual features into multi-dimensional embeddings to capture finer query–document relevance, and its successor PreFLMR [7] scales pre-training to over ten million image–text pairs, yielding a powerful multimodal retriever. Self-bootstrapped [ ] further proposes a co-training scheme in which a selector picks key knowledge documents for the answerer and the answerer returns pseudo-labels to refine the selector, bootstrapping both components without extra annotations.
>
> >Q6: Analysis on lambda
>
> Experimental results demonstrate that, within a reasonable range, the recall is insensitive to the choice of λ. This parameter weights the entropy-loss term: a larger λ encourages the predicted importance scores to approach a uniform distribution, whereas a smaller λ may drive the model to focus exclusively on the hardest instances, sacrificing sample diversity. We evaluated λ ∈ {0.005, 0.01, 0.1, 0.5} and observed that the re-ranking recall varied by less than 2%. Across all settings, entropy loss exhibited a consistent trend.

---

> > ### Author Response · Authors · 2025-11-21
> > **Reference**
> >
> > [1] Gui L, Wang B, Huang Q, et al. Kat: A knowledge augmented transformer for vision-and-language[C]//Proceedings of the 2022 conference of the North American chapter of the association for computational linguistics: human language technologies. 2022: 956-968.
> >
> > [2] Yuanze Lin, Yujia Xie, Dongdong Chen, Yichong Xu, Chenguang Zhu, and Lu Yuan. 2022. Revive: Regional visual representation matters in knowledge-based visual question answering. Advances in Neural Information Processing Systems, 35:10560–10571.
> >
> > [3] Lin W, Byrne B. Retrieval augmented visual question answering with outside knowledge[J]. arXiv preprint arXiv:2210.03809, 2022.
> >
> > [4] Qu C, Zamani H, Yang L, et al. Passage retrieval for outside-knowledge visual question answering[C]//Proceedings of the 44th International ACM SIGIR Conference on Research and Development in Information Retrieval. 2021: 1753-1757.
> >
> > [5] Gao F, Ping Q, Thattai G, et al. Transform-retrieve-generate: Natural language-centric outside-knowledge visual question answering[C]//Proceedings of the IEEE/CVF conference on computer vision and pattern recognition. 2022: 5067-5077.
> >
> > [6] Lin W, Chen J, Mei J, et al. Fine-grained late-interaction multi-modal retrieval for retrieval augmented visual question answering[J]. Advances in Neural Information Processing Systems, 2023, 36: 22820-22840.
> >
> > [7] Lin W, Chen J, Mei J, et al. Fine-grained late-interaction multi-modal retrieval for retrieval augmented visual question answering[J]. Advances in Neural Information Processing Systems, 2023, 36: 22820-22840.
> >
> > [8] Hao D, Wang Q, Guo L, et al. Self-bootstrapped visual-language model for knowledge selection and question answering[J]. arXiv preprint arXiv:2404.13947, 2024.

---

> > ### Comment · Reviewer_Nd2P · 2025-11-23
> >
> > Thanks for the response. If this paper is accepted, please make sure that clear results regarding Q2, Q6 will be included, along with the extended discussion of previous works in other KB-VQA datasets.
> >
> > I decided to slightly increase the score.

---

> > > ### Author Response · Authors · 2025-11-24
> > > **Response to Reviewer Nd2P**
> > >
> > > Thank you once again for your invaluable feedback, and we sincerely appreciate your willingness to engage with us and to raise the score. Regardless of the final acceptance decision, we will revise and strengthen our submission in full accordance with your suggestions！

---

### Official Review · Reviewer_GUxB · 2025-10-26

**Soundness:** 1
**Presentation:** 2
**Contribution:** 2
**Rating:** 4
**Confidence:** 3

**Summary:**

This paper proposes a three-stage framework called Adv-CL, which aims to improve the reliability and precision of KBVQA. The framework consists of three main components: Query-guided Multi-grained Recalling, Adversarial Re-ranker Training and Guarded Answer Generation.
Experiments on E-VQA and InfoSeek show that Adv-CL achieves state-of-the-art performance without fine-tuning large language models.

**Strengths:**

1. Training with a small model has relatively low costs.
2. GAG enhances the safety of the method.
3. Clear description of the method.

**Weaknesses:**

The methods seem to have no major issues, but there are several severe problems in the experiment.
1. The authors conduct experiments on E-VQA and InfoSeek and they provided the details of the InfoSeek in A.1.1. As described, the InfoSeek dataset consists of a training set and three evaluation sets. The authors did not specify which evaluation set they used for evaluation. It seems reasonable to report the results of each evaluation set separately.
2. ReflectiVA reports two settings on InfoSeek (28.3 and 40.1). Please clarify why your table uses 28.3 while ignoring 40.1?
3. For mR2AG, they conducted experiments on each evaluation set of InfoSeek. Why did the authors only use the worst-performing InfoSeek-Human as the baseline result? As baselines, both ReflectiVA and EchoSight claim their results are from InfoSeek's validation set. It seems the authors should use mR2AG's validation set results as the baseline instead of results of InfoSeek-Human.
4. The ablation study was conducted incompletely. The integration between the three modules in this article is not tight, so performing ablation on the full dataset should be relatively straightforward. An experimental setup similar to the main experiment is expected.
5. The case study is too brief. It would be better for the examples to demonstrate the complete workflow, including adversarial training and GAG.

Minor issues:
- Citation Format: arXiv:2411.15041,2024a. & arXiv:2411.15041,2024b.
- Typos: "repectively", A.1.4.

**Questions:**

See Weaknesses.

---

> ### Author Response · Authors · 2025-11-21
> **Response to Reviewer GUxB**
>
> Thank you very much for your detailed and insightful questions. Below are our responses.
>
> > Q1:Test dataset for InfoSeek
>
> Following prior work for a consistent comparison, we use the **validation** split of InfoSeek as our test set. We will make this point explicit in the paper.
>
> >Q2 and Q3: Result for ReflectVA and mr2ag
>
> - To ensure a fair comparison, we use the **visual retrieval results** as our baseline, as we also rely on visual candidates for entity retrieval.
> - We will update the InfoSeek results for mr2ag in the script.
>
> >Q3: Ablation study
>
> We measured the computation, final score, and latency introduced by QMR, ART, and GAG during training and inference, as shown in Table 1. Removing either QMR or ART causes a noticeable drop in performance, while their combined use achieves the best result: QMR supplies ART with a broad candidate set and high-quality sample pool, enabling more effective adversarial contrastive learning.
>
> >Q4:Case study
>
> We will follow your suggestion and add this part to the appendix.

---

> > ### Comment · Reviewer_GUxB · 2025-11-25
> >
> > Thank you for the rebuttal. However, several issues remain unclear to me.
> > ﻿
> > 1. **Inconsistency in the ablation results.** In the newly provided ablation table, the VQA score for the setting with all modules enabled is reported as 45.9, which differs from the 42.8 reported for w/ ART in the original submission. The value 42.8 does not appear anywhere in the new table. Could the authors clarify what caused this discrepancy? Was the model retrained, or was there an error in the initial or updated reporting?
> > ﻿
> > 2. **Missing revisions.** The authors stated that two parts of the paper would be revised. As of the time I am writing this new comment, I still do not see any visible changes in the manuscript. Could the authors confirm whether the revisions have actually been made?
> > ﻿
> > 3. **Concerns regarding mR2AG performance.** Since the authors mention updating the mR2AG results, and as reviewer iMhz pointed out, mR2AG outperforms the proposed method on both INFOSEEK and E-VQA, it remains unclear how the proposed method achieves any performance advantage. The rebuttal does not adequately address this concern.
> > ﻿
> > Given these unresolved issues, I am inclined to maintain my current score (with only adjustment possible in extreme cases).

---

> ### Author Response · Authors · 2025-11-26
> **Response to Reviewer GUxB**
>
> Thank you very much for your willingness to raise further questions and engage in discussion. Below are our **point-by-point** responses:
>
> > Q1. Apparent inconsistency in the ablation results
>
> We followed the reviewers’ common suggestion and added a **global ablation** in which each component is removed one at a time. Please note that the **42.8** in the original submission is the gain obtained when **only ART** is inserted (i.e., no QMR, no GAG). The gap between this number and the 45.9 reported for the full framework clearly shows the extra lift provided by QMR. In the new global-ablation table we no longer **isolate ART alone**; instead we report the performance drop when one component is removed while the other two are kept. Consequently, **there is no inconsistency**—the 45.9 can still be found in the main-results table of the original submission.
>
> > Q2. Location of the updates
>
> **To accelerate the review cycle we first answered every concern in the rebuttal comments, where all issues have been addressed in full detail.** The final PDF will be updated only after the discussion phase, incorporating all valuable suggestions in a single pass.
>
> > Q3. Comparison with mR²AG
>
> - As already explained to Reviewer imhz, mR²AG’s higher EVQA score is overwhelmingly due to its **commercial Google-Lens retriever**, whereas we use a pure open-weight CLIP+FAISS pipeline. Section 3 **(“Observation”)** shows that **the retrieval ceiling is currently the dominant bottleneck**; comparing two methods without aligning their retrievers is therefore unfair.
> - On InfoSeek, the comparison is further complicated by training scope: **mR²AG fine-tunes the entire LLM on ∼1.5 M examples (InfoSeek + Enc-VQA + NQ, ≈ 1.7× our instruction data).**
> We only train on Enc-VQA and treat InfoSeek as **zero-shot (see Appendix A.1.1).**
>
> We therefore regard our current results on InfoSeek—obtained under zero-shot conditions—as encouraging. Although we debated whether to include mR²AG as a main-table baseline owing to the aforementioned disparities, we ultimately decided to retain it in salute to this excellent work. In short, despite divergent training paradigms and budgets, the gains delivered by our approach remain substantial and meaningfully comparable. In the revision, we will move the comparison with mR²AG to the “Related Work” section only, citing it as an important reference rather than a head-to-head baseline.
>
> We sincerely appreciate your constructive feedback and careful reading, and we look forward to continuing the conversation to address any remaining concerns!

---

### Official Review · Reviewer_ESKh · 2025-10-31

**Soundness:** 2
**Presentation:** 2
**Contribution:** 2
**Rating:** 2
**Confidence:** 4

**Summary:**

This paper addresses the challenge of accurate knowledge retrieval in Knowledge-based Visual Question Answering (KBVQA), where existing systems often feed incorrect facts into the answer generator. To improve both recall quality and reranking discrimination, the authors propose the Adversarial Curriculum Learning (Adv-CL) framework, composed of three components: (1) Query-guided Multi-grained Recalling (QMR), (2) Adversarial Reranker Training (ART), and (3) Guarded Answer Generation (GAG).  Experiments on two KBVQA benchmarks demonstrate improved performance.

**Strengths:**

1. The paper provides a new framework that integrates multiple stages (recall, reranking, and answer generation) into a unified training paradigm.
2. The experiments cover two public KBVQA benchmarks and show the incremental effects over the current approach.

**Weaknesses:**

1. The framework introduces three new modules (QMR, ART, and GAG) built on top of existing methods such as FAISS and EVA-CLIP. This significantly increases pipeline complexity and may amplify error propagation between stages. The benefit-to-complexity ratio is unclear.

2. While the full model achieves performance gains, it is not convincingly shown whether all three modules are necessary. A simplified or modular version might achieve comparable performance. The improvement margins in Table 1 are modest and not consistently significant (ps. the best result on E-VQA is mR2AG which should be bolded other than the proposed methods).

3. The paper does not isolate the specific contributions of the retriever and reranker components. It remains unclear how much each stage contributes to the final performance or whether the current paradigm is inherently limited by the capacity of the base models (i.e.,  LLM backbones).

4. I'd like to see computational comparison over existing approaches, which is misleading in the current scope.

**Questions:**

See above.

---

> ### Author Response · Authors · 2025-11-21
> **Response to Reviewer ESKh**
>
> Thank you very much for your time in reviewing our submission. Below are our responses.
>
> >Q1:Benefit-to-complexity
>
> The Retrieval-Reranking-Generation pipeline represents the typical flow for addressing Knowledge Base Visual Question Answering (KBVQA) problems. Prior work, such as [echosight], also adopted these three characteristic stages and achieved notable performance and progress on knowledge-intensive tasks.
> Specifically, the retrieval stage efficiently filters visually similar knowledge from a large-scale candidate pool by calling the FAISS library for rapid vector search; the encoding of candidate images is performed offline, which further minimizes online retrieval latency.
> EVA-CLIP acts as the feature extraction backbone. Due to the flexibility of our framework, it can be swapped for simpler or more lightweight backbones depending on deployment requirements (e.g., computational budgets).
> A reranking phase is subsequently introduced to refine the selection of evidential knowledge sections through the incorporation of multi-modal information—this aligns with the principle of coarse-to-fine filtering.
> The final, indispensable step involves augmenting the LLM's answer generation with the retrieved external knowledge paragraphs. In summary, these three stages are necessary within the pragmatic context of KBVQA. Furthermore, a core motivation of our method is to mitigate the potential issue of error propagation by proposing QMR to obtain multi-perspective candidate samples.
>
> >Q2 & Q3:Global Ablation
>
> The ablation study presented in Table 3, C, D in the paper experimentally confirms the necessity of each module, showing that a simplified version cannot be effectively obtained. Besides, we supply the global ablation and cost analysis in Table 1 from the response. Both the retriever and the re-ranker are indispensable. The query-guided multi-grained feature extraction is essential as it provides the re-ranker with a wide and diverse set of candidate samples, enabling the re-ranker to use adversarial training to learn hard negative examples that are otherwise difficult to distinguish. The bolding in Table 1 in the paper has been corrected.
>
> >Q4:Cost Analysis
>
> The computational cost and latency introduced by incorporating QMR, ART, and GAG during training and inference are presented in Table 1. Our improvements do not introduce a noticeable impact on the latency.
>
> ---
> As Reviewer iMhz and Reviewer Nd2P noted, the merit of our work is **not simply delivering a multi-step pipeline**, but rather: (i) pinpointing flaws in current KB-VQA systems through a systematic analysis of existing paradigms (Related Work) and empirical observations (Observation); (ii) employing QMR for dual-path, multi-granularity, multi-view recall to feed ART a stream of confusing, high-quality negatives; and (iii) using adversarial training to re-weight these negatives on the fly, guiding the reranker through a curriculum that steadily raises discrimination difficulty and thus its training quality.

---

### Official Review · Reviewer_iMhz · 2025-11-01

**Soundness:** 3
**Presentation:** 3
**Contribution:** 2
**Rating:** 4
**Confidence:** 3

**Summary:**

This paper identifies retrieval quality as the main bottleneck in knowledge-based VQA, by observing the significant gap between accuracies of a same generator with ground-truth knowledge vs. retrieved knowledge. To address this, authors propose Adv-CL which is a three-stage pipeline. At stage one, the method uses a VLM to select most relevant patches to the input image, query-guided multi-grained Recalling, combines global image features with local patch features to improve knowledge recall. At stage 2, Adversarial Reranker Training through a minimax optimization between a modulator and reranker, in which modulator tries to assign higher scores to the most challenging negative samples, while reranker learns to minimize the contrastive loss, facilitating learning from truly challenging negative examples. At stage 3, guarded answer generation mechanism, to assess reliability of the retrieved knowledge and enable abstention when evidence is not reliable. Evaluations on two datasets, shows that the proposed method achieves higher VQA accuracy and better recall quality than the state of the art.

**Strengths:**

- Importance of the problem & key issues: The authors have identified a critical bottleneck in KBVQA models, where poor retrieval
significantly deteriorates VQA performance.
- Paper's motivation is sound. The paper grounds its design in three empirical observations: retrieval–generation gap, negative‑signal decay, and factual contamination
- The proposed is a plug-and-play method with frozen VLM/LLMs, and does not require LLM fine-tuning.
- The evaluations demonstrate that the proposed method achieves higher performance compared to baselines, and the method performances are consistent across three different LLMs.

**Weaknesses:**

- The paper do not provide a cost analysis of the proposed method in terms of the retrieval latency, reranking cost, and end-to-end costs.
- Details of the GAG stage are not provided. For example, details of the prompt inspection and discriminator are missing (this is important for reproducibility). Additionally, AP/AR/VAR are reported, but the trade‑off curve (against threshold) and its impact on final accuracy under different abstention policies aren’t shown.
- In table 2, mR²AG shows a higher score (55.9) on E-VQA dataset, than the proposed. Additionally, the table mixes methods & generators, hence the baseline comparisons are not apple-to-apple comparisons. The paper should present the results in a way to facilitate comparison of methods on the same generator, for a fair comparison.
- Moderate novelty: Multi‑grained retrieval and dynamic hard‑negative mining are known ideas. Paper's novelty lies in (a) operationalizing query‑guided patch features for the retrieval stage and (b) casting negative weighting as an adversarial curriculum with an entropy‑regularized budgeted modulator. Additionally, GAG abstention mechanism is pragmatic rather than novel.

**Questions:**

- What is the cost and latencies of the method (end-to-end) and per component?
- How is the prompt inspection for GAG designed? Is the abstention decision a binary decision made by LLM, or does the LLM return an abstention score?
- Plot AP, AR, VAR trade-off curves across thresholds and show overall accuracy change under different abstention policies (e.g., fixed refusal budget).

---

> ### Author Response · Authors · 2025-11-21
> **Response to Reviewer iMhz (1)**
>
> Thank you very much for taking the time to review our submission and for your insightful suggestions. Below is our response to your concerns.
> > Q1:Cost analysis of the proposed method in terms of the retrieval latency, reranking cost, and end-to-end costs.
>
> We measured the computation and latency introduced by QMR, ART, and GAG during training and inference, as shown in Table 1.
> - The main latency in the QMR stage comes from computing the query-weighted features. In fact, our system already performs image-to-image retrieval, which, powered by Faiss, supports efficient large-scale search. The two-stage recall runs in parallel and therefore adds no noticeable delay. By contrast, mainstream alternatives that mix text and image modalities require a second encoding step. Although QMR incurs extra inference cost compared with pure image–image retrieval, its latency is on par with the dominant Image+Text retrieval paradigm, while additionally enabling multi-granularity retrieval fused with query information. This yields high-quality negatives for ART’s contrastive training, bringing substantial gains in recall quality.
> - ART does raise training costs because of the extra regulator, but the overhead is **tiny** for two reasons. First, the regulator’s input sequence length equals the number of positives and negatives in one batch—typically only a few dozen—and the regulator itself is just two standard Transformer blocks, so the added reranker-training latency is minimal. Second, ART is used only at training time and adds zero inference cost; it is a one-time expense that permanently benefits the RAG system, which we regard as worthwhile.
> - GAG is invoked only at RAG inference and offers users an optional behavior: when the model refuses to answer, given the current retrieval results, the user can choose either to continue retrieving or to stop waiting. It never proactively introduces extra inference latency.
>
> >Q2:Details of GAG
>
> We provide GAG with two optional strategies: a prompt-based inspection mechanism and a dedicated retrieval discriminator.
> - For the prompt-based inspection, the prompt is as follows:
> *\<Query>\n\<Context>\nPlease first determine whether the provided context can answer the question posed for the image, and output it in the format of\<think>Yes\</think> or \<think>No\</think>.
> If it is \<think>Yes\</think>, it means that the correct answer to the question can be obtained based on the provided context, and the answer to the question is output in the format of \<answer>\</answer>.
> If it is \<think>No\</think>, it means that the correct answer to the question cannot be obtained based on the provided context, and no other output is required.*
> - For the retrieval discriminator, we sampled 20K training instances from the E-VQA dataset. Each instance consists of a \<Query> and \<Context> pair and a binary label of Yes or No. If the \<Query> and \<Context> form a matched positive pair according to the dataset annotations, the corresponding label is Yes; otherwise, it is No. To improve training efficiency, we selected negative samples for each \<Query> exclusively from the hard set filtered by the retrieval module. Additionally, we balanced the ratio of samples corresponding to the two labels in the training data to prevent model bias.
> - The definitions and computational details of AP/AR/VAR are given in Appendix 1.3; these metrics are threshold-independent and better capture the gain contributed by the GAG component. **As illustrated in Figure 1(c), relying on the final VQA score to gauge GAG’s contribution is unreliable, since semantically similar yet incorrect answers are not only misleading but can also receive higher VQA scores, undermining the system**. We therefore introduce a safeguard that enables the model to refuse answering when the retrieved context is irrelevant, preventing the generation of harmful misinformation. Ablation studies in Appendix 2.2 verify GAG’s effectiveness, and we additionally report its ablation results measured by the final VQA score for reference in Table 1.

---

> ### Author Response · Authors · 2025-11-21
> **Response to Reviewer iMhz (2)**
>
> >Q3: Results of mr2ag on E-VQA and results on different generators
>
> mR²AG’s high score on E-VQA primarily stems from its Google-Lens retriever—a commercial system that integrates OCR, visual entity recognition, and Google’s knowledge graph—yielding a significantly higher recall ceiling than pure CLIP-based vector retrieval. Our approach employs open-source EVA-CLIP-8B + FAISS for embedding-based retrieval,  thus is inherently weaker in retrieval capacity. Moreover, mR²AG fine-tunes its generator extensively, whereas our framework uses frozen LLMs in a plug-and-play manner—making the performance gap reasonable. Table 2 follows the KBVQA standard by retaining each method’s original generator to avoid confounding “method” and “generator” variables. To disentangle these factors, we have provided results with three frozen generators (Mistral-7B, LLaMA-3-8B, Qwen2.5-7B) for comparison with prior work. Following your suggestion, we further supplement the full pipeline results using Vicuna-7B (the model used in RORA-VLM and other baselines) as follows: 46.3 for E-VQA and 34.0，33.8，33.9 for InfoSeek.
>
> >Q4:Innovation
>
> Although multi-grained retrieval has been exploited before, its role in this work is not a simple plug-in but a mechanistic extension tightly coupled with hard-negative mining. Query-guided Multi-grained Recalling (QMR) decomposes an image into multiple regional views whose embeddings are independently queried against the knowledge base. Consequently, a picture whose dominant tag is “building” can retrieve documents about “river” or “mountain” if those objects co-occur in the background, yielding a thematically mixed candidate pool. This pool intrinsically supplies high-confusion negatives for the subsequent adversarial re-ranker, shifting the generation of hard negatives upstream from the training loop to the retrieval stage. As a result, contrastive learning is no longer constrained by static or random sampling; instead, it operates on a curriculum of continuously escalating semantic difficulty, playing a minimax game against the discriminator. The process mirrors human curricula: the model progressively refines its decision boundary by distinguishing increasingly ambiguous boundary cases rather than by naïvely stitching together retrieval and ranking components.

---

### Author Response · Authors · 2025-11-21
**Response to Cost Analysis and Global Ablation**

**Table 1: Cost Analysis and Global Ablation**

| QMR | ART | GAG | GFLOPS | TSpeed (it/s) | Latency(ms) | VQA Score |
| :---: | :---: | :---: | :---: | :---: | :---: | :---: |
| - | - | - | 14,334 | 2.12 | 2,793 | 41.8 |
| $\checkmark$ | - | - | 14,360 | 2.12 | 3,744 | 42.3 |
| $\checkmark$ | $\checkmark$ | - | 14,360 | 1.90 | 3,744 | 46.0|
| $\checkmark$ | $\checkmark$ | $\checkmark$ | 15,070 | 1.90 | 3,744 | 45.9 |
| - | $\checkmark$ | $\checkmark$ | 15,044 | 1.90 | 2,793 | 42.5 |
| $\checkmark$ | - | $\checkmark$ | 15,070 | 2.12 | 3,744 | 42.1|

GFLOPS quantifies the total computational complexity of the model. TSpeed (it/s), representing Training Speed, is defined such that a larger numerical value corresponds to faster training time per iteration. Latency (ms) measures the millisecond-level delay experienced during the model's inference process.

---

### Author Response · Authors · 2025-12-03
**Rebuttal Summary**

Dear Area Chair and Reviewers,

We sincerely appreciate the valuable and constructive feedback provided by all reviewers. We have utilized this rebuttal period to conduct substantive revisions and detailed clarifications in response to the received comments. We believe the revised manuscript is significantly improved in terms of technical rigor and experimental completeness.

Our main clarifications and revisions focused on the following key areas:

**I. Response to Global Ablation and Cost Analysis**

(Reviewers iMhz, ESKh, GUxB, and Nd2P)

Reviewers universally inquired about the efficiency and necessity of our core components. We have now provided comprehensive ablation studies for all three key components and delineated a detailed analysis of their impact on latency, training speed, and GFLOPs (See Table 1 in our comments). The results confirm that our method provides a clear uplift in Recall and VQA score with an acceptable, marginal increase in computational overhead.

**II. Defense of Novelty and Core Mechanism**

(Reviewers iMhz and Nd2P)

We appreciate the divided feedback on our paper's novelty (Reviewers ESKh and GUxB acknowledged it, while iMhz and Nd2P expressed skepticism). We wish to reaffirm that the core innovation of our work lies in the Adaptive Hard Negative Mining Mechanism within our ART framework. While the concept of hard negatives is not new, our experiments conclusively demonstrate that static sampling strategies fail to keep pace with training, leading to premature exhaustion of the ranking model’s supervisory signal. Our proposed Dynamic Hard Negative Mining mechanism actively and gradually increases negative sample difficulty as training progresses, continually enhancing the reranker's ability to distinguish genuine evidence.

**III. Clarifications on Specific Issues**

We have addressed all remaining specific comments through direct explanations and new additions to the manuscript:

**m2AG Experimental Results:** We have provided necessary explanations and corrections regarding the m2AG experimental outcomes in the OpenReview discussion threads (https://openreview.net/forum?id=VAgRyr8R5t&noteId=yGAomFSoBv and https://openreview.net/forum?id=VAgRyr8R5t&noteId=vvo9AWoxog).

**GAG Details and Avoidance Strategies**: We added details regarding the GAG mechanism and the impact of different avoidance strategies on accuracy in Appendix A.1.5 and Table E (R# iMhz).

**Gain vs. Complexity Trade-off**: We provided an explanation of the benefit-to-complexity ratio in the discussion thread (R# ESKh）

**Dataset Details and Case Study**: We have added explicit details regarding our experimental datasets in Appendix A.1.1 and, following Reviewer GUxB's suggestion, included a complete pipeline Case Study in Appendix A.4.

**Dataset Selection and Sensitivity**: We have clarified the rationale behind our current dataset choices (in the discussion thread) and incorporated related work concerning Open-domain RAG datasets (e.g., OK-VQA) in Appendix A.3. Furthermore, we added a sensitivity analysis for the hyperparameter $\lambda$ in Appendix A.2.4 (R# Nd2P).

**Conclusion**

Following these substantive revisions and detailed clarifications, we firmly believe the current version of **the manuscript has successfully mitigated all core reviewer concerns**. We are particularly pleased to note that Reviewer Nd2P has confirmed that **our Rebuttal addresses his primary concerns, including the common issues of novelty, ablation completeness, and computational cost**. We have also provided **further constructive clarification** in response to the follow-up questions raised by Reviewer GUxB. We respectfully request the Area Chair to consider our **detailed responses, newly added analysis, and experimental work** against the original reviews (including those from the unresponded Reviewers iMhz and ESKh) in their comprehensive assessment of our paper.

Thank you for your time and further consideration.

---

### Note · Authors · 2026-01-06

I have read and agree with the venue's withdrawal policy on behalf of myself and my co-authors.